# Forecasting Thailand's mobility trends using Feature Engineered XGBoost for pandemic crisis movement management

Aritath Siraphatwongkorn[1☯], Thanin Methiyothin[2‡], Kittisak Onuean[3‡],
Krisana Chinnasarn[4‡], Athita Onuean[5‡], Insung Ahn[6☯*], Suwanna Rasmequan[7☯*]

**1** Faculty of Informatics, Burapha University, Chonburi, Thailand, **2** Faculty of Informatics, Burapha University, Chonburi, Thailand, **3** Faculty of Science and Social Sciences, Burapha University, Sakaeo, Thailand, **4** Faculty of Informatics, Burapha University, Chonburi, Thailand, **5** Faculty of Informatics, Burapha University, Chonburi, Thailand, **6** Korea Institute of Science and Technology Information, Yuseong-gu, Daejeon, Republic of Korea, **7** Faculty of Informatics, Burapha University, Chonburi, Thailand

☯ These authors contributed equally to this work.
‡ These authors also contributed equally to this work.
\* rsuwanna@buu.ac.th (SR); isahn@kisti.re.kr (IA)

## Abstract

The COVID-19 pandemic significantly disrupted global mobility patterns, with widespread population movement playing a key role in the transmission of the virus. In such a situation, Google introduced the Community Mobility Reports, which use anonymized and aggregated location data to monitor changes in movement across various location categories. These mobility trends provide important insights that help inform timely public health interventions and support data-driven decisions during and after the pandemic. This study aims to forecast human mobility trends in Thailand during the COVID-19 pandemic using data from Google's reports. Three forecasting models were applied: Facebook Prophet, ARIMA, and Feature Engineered XGBoost. The Granger Causality Test was used to examine the relationship between mobility patterns and COVID-19 case numbers across different phases of lockdown. The results indicated that Feature Engineered XGBoost demonstrated the highest overall accuracy in forecasting mobility trends across all six location categories. In conclusion, this study demonstrates the effectiveness of machine learning models in forecasting mobility movement across various location types while public health restrictions have been implemented. This underscores the importance of understanding mobility patterns as a key factor in disease transmission. The insights gained from this analysis can help formulate strategic and targeted mobility management policies and public health responses for future outbreaks, ultimately helping to contain the spread of disease more effectively.

**Data availability statement:** The data underlying the results presented in the study are available from https://github.com/apisitgo/COVID19-Mobility-Forecasting-TH or you can directly access via the following address: 1. https://www.google.com/covid19/mobility/ 2. https://aida.informatics.buu.ac.th/ 3. https://www.bsg.ox.ac.uk/research/covid-19-government-response-tracker.

**Funding:** This research was supported by the Government-wide R&D to Advance Infectious Disease Prevention and Control, Republic of Korea (grant number: RS-2023-KH140419). This research was also co-funding by the Faculty of Informatics, Burapha University, Thailand. The funders had no role in study design, data collection and analysis, decision to publish, or preparation of the manuscript.

**Competing interests:** The authors have declared that no competing interests exist.

## 1. Introduction

The global outbreak of Coronavirus Disease 2019 (COVID-19) has profoundly reshaped human mobility patterns across countries and regions [1]. Since the virus is primarily transmitted through close human contact, understanding population movement is essential for managing and forecasting the spread of the disease. In response to the pandemic, governments worldwide implemented a variety of public health measures—such as travel restrictions, lockdowns, and social distancing policies—to curb transmission rates [2–4]. These interventions significantly influenced how people traveled, worked, and interacted within their communities, leading to visible and measurable changes in mobility behaviors [5]. Empirical evidence further demonstrates that restrictions on human movement can substantially alter transmission dynamics; for example, Wang et al. [6] showed that mobility reductions in China effectively disrupted the underlying COVID-19 transmission network, highlighting the strong association between mobility flows and epidemic spread.

To support research and policymaking during the crisis, Google launched the COVID-19 Community Mobility Reports (GCMR) [7]. These reports offer anonymized, aggregated mobility data across six key categories: retail and recreation, grocery and pharmacy, parks, transit stations, workplaces, and residential areas. By comparing current mobility levels to a pre-pandemic baseline, GCMR enables researchers to monitor real-time behavioral responses to public health interventions [8,9]. Studies in other regions, such as the United States, have similarly documented substantial shifts in mobility across these domains, reinforcing the relevance of mobility data in understanding population behavior during health emergencies [10]. Moreover, research from Japan has shown that human mobility and regional connectivity patterns can shape the timing and magnitude of epidemic waves, underscoring the importance of incorporating mobility indicators into models designed to forecast disease trends and inform policy responses [11].

In Thailand, four major waves of COVID-19 outbreaks have been recorded: the Wuhan wave (February 4 – December 14, 2020), Alpha-Beta wave (December 15, 2020 – March 31, 2021), Delta wave (April 1 – December 31, 2021), and Omicron wave (January 1 – October 15, 2022) [12] as shown in Table 1. Each wave differed in terms of infection rates, severity, and government response, resulting in distinct mobility patterns during each period.

To contain rising case numbers, the Thai government introduced a series of policy interventions, including various levels of lockdowns and travel restrictions as shown in Table 2 [13]. This study classifies these measures into three levels of stringency. Thailand provides a compelling case for examining the dynamic interplay between public health policy, epidemic severity, and behavioral adaptation, given its evolving and context-specific responses to each outbreak. Accurate forecasting of mobility trends during such public health emergencies is critical for timely decision-making and the efficient allocation of healthcare and transportation resources [14].

While previous studies [18–20] have utilized time-series methods to predict mobility patterns, comparative analyses of multiple forecasting models specific to the COVID-19 context remain relatively limited as to the best of our knowledge. This

**Table 1. The different periods of COVID-19 outbreaks in Thailand.**

| Periods | Period name | Time period |
|---|---|---|
| 1 | First wave (Wuhan) | February 4 – December 14, 2020 |
| 2 | Second wave (Alpha-Beta) | December 15, 2020 – March 31, 2021 |
| 3 | April 2021 wave (Delta) | April 1 – December 31, 2021 |
| 4 | January 2022 wave (Omicron) | January 1 – October 15, 2022 |

**Table 2. Levels of COVID-19 lockdown measures in Thailand.**

| Lockdown Level | Approximate Period | Description of Measures | Reasons for Implementation |
|---|---|---|---|
| Strict Lockdown (High Restrictions) [15–17] | Mar. 26 – Apr. 30, 2020 Jul. 12 – Aug. 31, 2021 | - Curfew from 9:00 PM to 4:00 AM<br>- Closure of malls (only supermarkets/pharmacies open)<br>- Inter-provincial travel ban<br>- 100% work-from-home<br>- No gatherings of more than 5 people | - Daily cases exceeded 10,000<br>- Delta variant outbreak<br>- Overwhelmed healthcare system |
| Moderate Lockdown (Moderate Restrictions) [15,17] | May. 1 – Jun. 30, 2020 Sep. 1 – Sep. 30, 2021 | - Closure of entertainment venues and schools<br>- Limited mall operating hours<br>- Work-from-home encouraged<br>- Gatherings limited to 20–50 people | - Moderate outbreak<br>- Balance needed between public health and economic recovery |
| Soft Lockdown (Mild Restrictions) [16,17] | Jan. 1 – Jan. 31, 2021 | - Businesses open with health measures (DMHTT)<br>- Inter-provincial travel allowed<br>- Work-from-home encouraged but not enforced | - Localized outbreaks<br>- Focus on targeted control<br>- Policy shift toward "Living with COVID" |

study seeks to address this gap by assessing the performance of three forecasting approaches —Facebook Prophet, ARIMA, and Feature Engineered XGBoost—using GCMR data of Thailand. In this study, we also examine the influence of external factors such as daily confirmed COVID-19 cases and policy stringency on mobility patterns. Furthermore, the study investigates the potential causal relationship between mobility and subsequent changes in COVID-19 case numbers using Granger causality analysis. The following research questions were formulated to guide the study.

**Research question:**

RQ1: Do mobility trends from Google Community Mobility Reports (GCMR) and government policy measures influence the number of confirmed COVID-19 cases in Thailand, and if so, to what extent?

RQ2: Can mobility data collected during a pandemic be utilized to model and forecast population movement patterns that support the implementation of appropriate government interventions to limit disease transmission?

This paper is organized into four main sections. The introduction section presents the background and context of the study. The materials and methods section describes the data sources, analytical procedures, forecasting models, and evaluation metrics employed in this research. The results section reports and analyzes the study's findings. Finally, the conclusion section summarizes the key results, their implications, and discusses possible future work.

## 2. Materials and methods

### 2.1. Datasets and preprocessing

Datasets declaration statement: In this study, the data collection and analysis method complied with the terms and conditions of the source of the data.

In this study, we use daily mobility data in Thailand retrieved from the Google Community Mobility Reports or GCMR [7], covering the period from February 15, 2020, to December 31, 2022. The dataset includes six mobility categories, as listed in Table 3.

In this dataset, there is a 5-weeks baseline data starting from January 3 – February 6, 2020. The example of baseline value used to measure the change in mobility movement across 6-different sectors during the COVID-19 pandemic are shown in Table 4.

We analyzed the GCMR datasets by plotting a trend line graph as shown in Fig 1. The "Green line" is used to show the changing of mobility movement during the pandemic period. We then added the information of level of Governmental intervention in Thailand using the Stringency Index from the Oxford COVID-19 Government Response Tracker (OxCGRT) indicated by "Blue line" [21]. This composite index, ranging from 0 to 100, summarizes the strictness of public health policies by aggregating eight policy dimensions: school closures, workplace closures, cancellation of public events, gathering restrictions, public transport closures, stay-at-home requirements, internal movement restrictions, and international travel controls. This index provided a quantifiable measure of government response over time.

Additionally, epidemiological data on daily confirmed COVID-19 cases per million people in Thailand retrieved from the COVID-19 data repository maintained by the Johns Hopkins University Center for Systems Science and Engineering or JHU-CSSE [22] has been used as one of the factors for our analysis indicated by "Red line". This epidemiological data has been renowned and was used in different projects. We also previously include this data in our platform called

**Table 3. Definitions of mobility categories in the Google Community Mobility Reports.**

| No. | Category | Definition | Examples |
|---|---|---|---|
| 1 | Retail and Recreation | Refers to places related to leisure, entertainment, and non-essential shopping activities. | Restaurants, cafes, shopping malls, movie theaters, museums |
| 2 | Grocery and Pharmacy | Includes locations where people purchase essential food, household items, and medical supplies. | Supermarkets, grocery stores, food markets, convenience stores, pharmacies |
| 3 | Parks | Covers outdoor public spaces used for recreation and physical activity. | Public parks, beaches, botanical gardens, national parks, outdoor plazas |
| 4 | Transit Stations | Refers to transportation hubs used for public commuting. | Subway stations, bus terminals, train stations, ferry ports |
| 5 | Workplaces | Indicates places where people go to perform work-related tasks. | Offices, business centers, factories, government buildings |
| 6 | Residential areas | Represents time spent in places of residence, often associated with staying at home. | Private homes, apartments, residential neighborhoods |

**Table 4. Example of daily mobility in Thailand derived from Google Community Mobility Reports.**

| Date | retail_and_recreation_ percent_change _from_baseline | grocery_and_pharmacy_ percent_change _from_baseline | parks_percent_ change_from_ baseline | transit_stations_ percent_change_ from_baseline | workplaces_ percent_change_ from_baseline | residential_ percent_change_ from_baseline |
|---|---|---|---|---|---|---|
| 6/3/2020 | −5 | 2 | −9 | −11 | −6 | 4 |
| 7/3/2020 | −6 | 1 | −16 | −13 | 2 | 3 |
| 8/3/2020 | −7 | 1 | −11 | −15 | 1 | 2 |
| 9/3/2020 | −4 | 1 | −8 | −13 | −6 | 4 |
| 10/3/2020 | −3 | 1 | −8 | −13 | −7 | 4 |
| 11/3/2020 | −4 | 0 | −8 | −15 | −7 | 4 |
| 12/3/2020 | −6 | −1 | −9 | −15 | −7 | 4 |

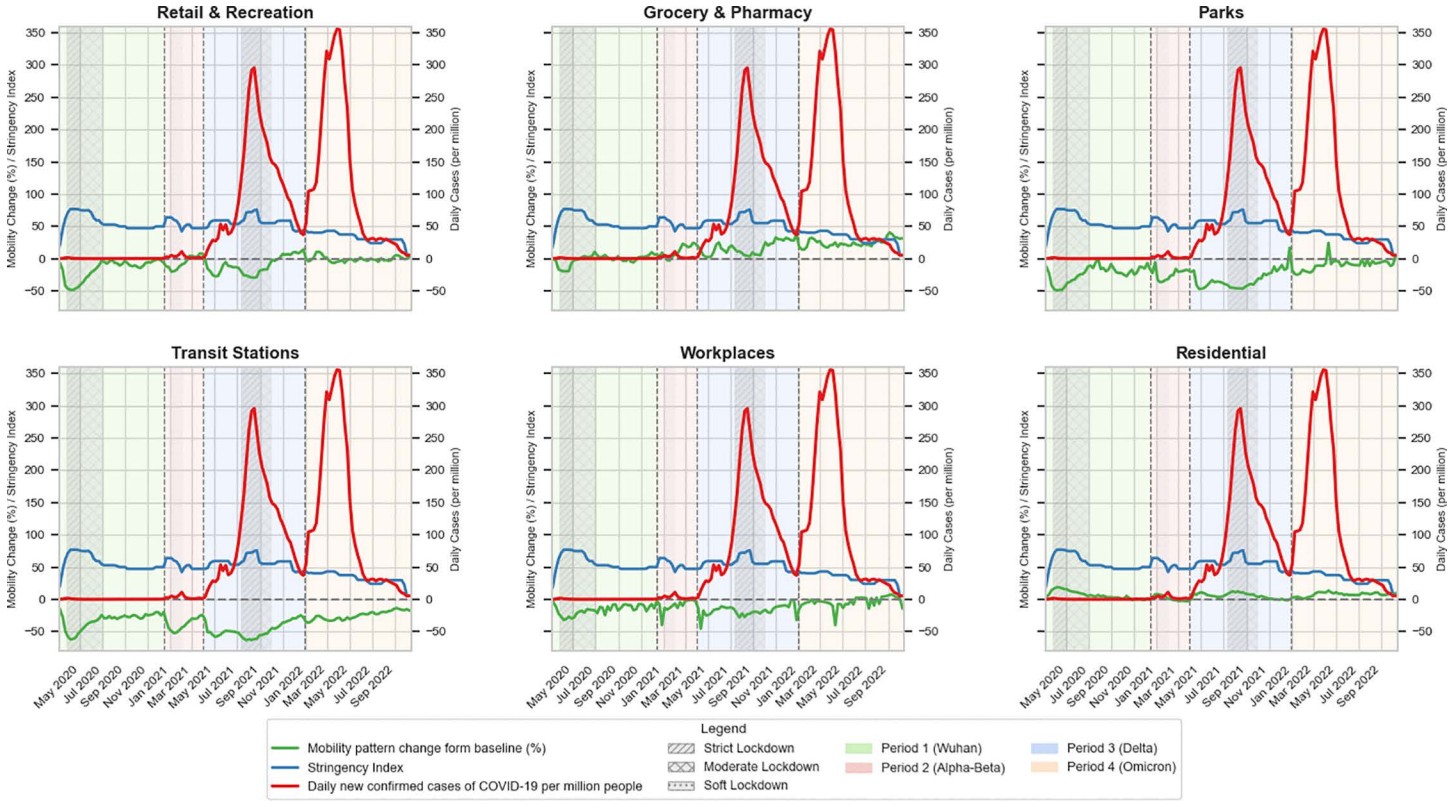

**Fig 1. The analysis of trends in mobility, policy stringency, and COVID-19 case rates across different location categories during the period of 2020–2022, considering the various phases of COVID-19 and government implications.**

AIDA platform [23] developed at our institute, which compiles open-access pandemic datasets relevant for public health research in Thailand.

We integrated all the three datasets mentioned above using "Date" as a primary key to ensure consistency in temporal granularity. The final dataset was partitioned into training and testing subsets at a 90:10 ratio. We decide to use this ratio to ensure that the models had sufficient historical data to capture underlying mobility patterns, particularly during the dynamic and rapidly changing circumstances of the COVID-19 pandemic. Given the limited data availability for certain phases of the outbreak and the high variability in human behavior due to sudden policy changes, a 90:10 split helped preserve enough sequential data for training while maintaining a representative and temporally relevant subset for performance evaluation. This strategy choice aligns with the best practices in time series forecasting, particularly when the data exhibits non-stationarity and is influenced by external shocks [24,25]. The temporal range of the data extends from the year 2020 to the year 2022, thereby facilitating a thorough examination of mobility trends throughout the pandemic.

The multi-series line graph mentioned above in Fig 1 illustrates trends in mobility (green), policy stringency (blue), and daily new confirmed COVID-19 cases per million (red) across six location categories: Retail & Recreation, Grocery & Pharmacy, Parks, Transit Stations, Workplaces, and Residential for the time span from 2020 to 2022. Shaded areas denote distinct pandemic phases: Period 1 (Wuhan), Period 2 (Alpha-Beta), Period 3 (Delta), and Period 4 (Omicron), with additional pattern style indicating the severity of lockdown measures (Strict, Moderate, Soft). Mobility trends represent changes in visitation relative to baseline levels, while the stringency index quantifies the rigor of governmental response

measures. Clear surges in COVID-19 cases, particularly during the Delta and Omicron waves, are accompanied by notable shifts in both mobility patterns and policy stringency.

To address the first research question no. 1 —"Do mobility trends from Google Community Mobility Reports (GCMR) and government policy measures influence the number of confirmed COVID-19 cases in Thailand, and if so, to what extent?"— this study divided the pandemic timeline into nine distinct periods, each reflecting a specific level of government intervention. These periods range from the initial phase with no formal response to progressively stricter lockdown measures, classified as soft, moderate, and strict. This segmentation is crucial for capturing the dynamic interactions between mobility and infection rates within varying policy contexts. Government interventions directly shape human behavior and mobility patterns, which subsequently influence virus transmission dynamics. By analyzing mobility and case data within these well-defined policy phases, the study enables a more precise and context-sensitive assessment of their causal relationships. The classification of these periods is grounded in official policy actions and implementation timelines, as detailed in Table 5.

The results of the Granger Causality analysis, as shown in Fig 2, indicate that the level of lockdown measures had a significant impact on the relationship between population mobility and daily confirmed COVID-19 cases as follows:

The Granger Causality results obviously showed that during the strict lockdown periods (Periods 2 and 7) in Thailand, mobility categories such as Retail & Recreation and Grocery & Pharmacy demonstrated strong causal relationships with daily COVID-19 cases. This reflects the necessity-driven behavior of the Thai population, who, despite facing stringent restrictions, still needed to leave their homes for essential supplies such as food and medicine. Notably, Parks consistently showed significant causal effects during these periods, highlighting their role as perceived safe outdoor spaces where people could engage in physical activity or seek mental relief. Given the closure of indoor recreational venues such as malls, gyms, and restaurants, parks became important outlets for the population to cope with stress and maintain well-being during times of severe restrictions.

In the moderate lockdown periods (Periods 3 and 8), the highest number of significant associations across various mobility categories was observed, particularly in Transit Stations, Retail & Recreation, and Residential Areas. This pattern suggests a partial recovery of economic and social activities following the relaxation of government measures, while public caution remained high. Importantly, Workplaces and Parks continued to exhibit significant causal relationships, reflecting the Thai government's allowance for certain businesses, such as factories and essential services, to resume operations. Consequently, workplaces and parks remained critical points of interaction and potential virus transmission during these intermediate phases of restriction.

During the soft lockdown period (Period 5), the overall impact on mobility categories became more limited, with only Grocery & Pharmacy showing a significant causal relationship with daily cases. This indicates that while restrictions were minimal, essential daily activities such as grocery shopping remained unavoidable. Meanwhile, reduced causal

**Table 5. COVID-19 control phases in Thailand and corresponding government interventions.**

| Period | Timeline/ Date Range | Lockdown Level |
|---|---|---|
| Period 1 | 6 March – 25 March 2020 | |
| Period 2 | 26 March – 30 April 2020 | Strict Lockdown |
| Period 3 | 1 May – 30 June 2020 | Moderate Lockdown |
| Period 4 | 1 July – 31 December 2020 | |
| Period 5 | 1 January – 31 January 2021 | Soft Lockdown |
| Period 6 | 1 February – 11 July 2021 | |
| Period 7 | 12 July – 31 August 2021 | Strict Lockdown |
| Period 8 | 1 September – 30 September 2021 | Moderate Lockdown |
| Period 9 | 1 October – 15 October 2022 | |

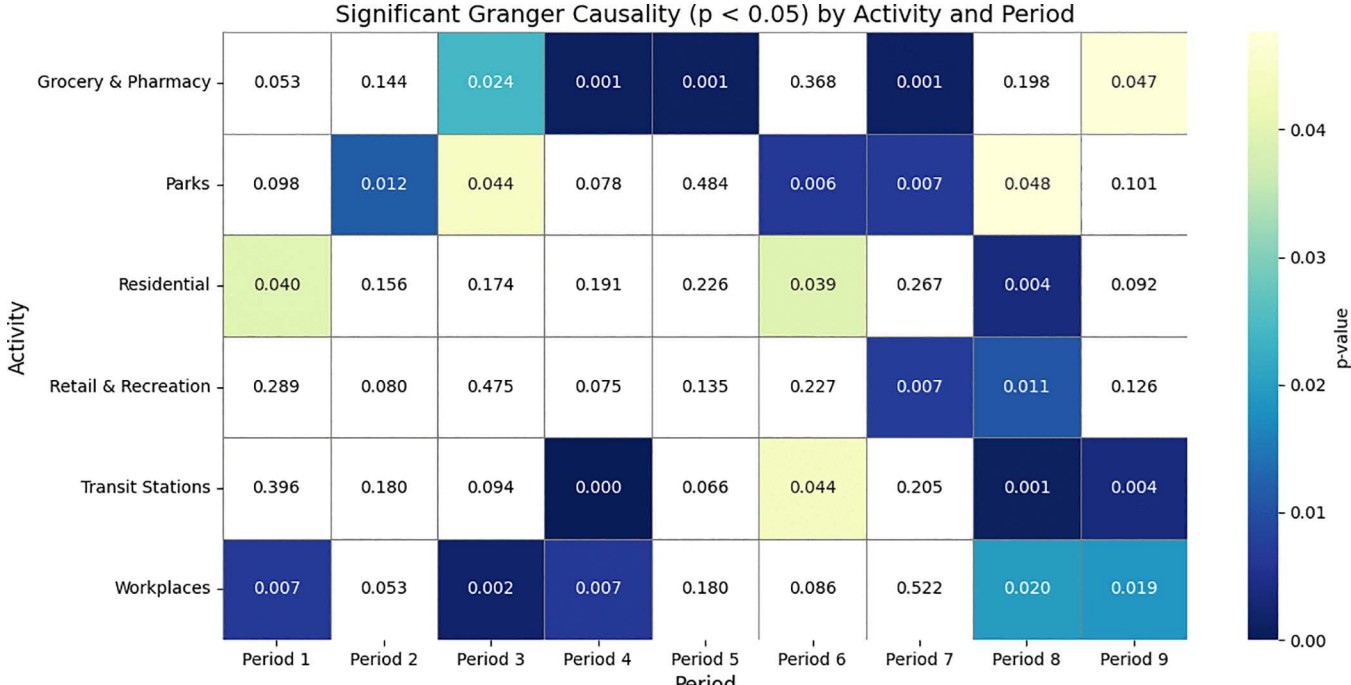

**Fig 2. Granger causality (p < 0.05) across activities and time periods with government intervention.**

relationships in other categories suggest a degree of voluntary risk avoidance by the Thai public, who, despite the absence of strict enforcement, continued to self-regulate their mobility behaviors due to ongoing pandemic awareness.

Finally, during the no-lockdown periods (Periods 1, 4, 6, and 9), strong causal links persisted in categories such as Residential Areas, Parks, and Grocery & Pharmacy. Notably, Workplaces and Transit Stations exhibited consistent and significant causal effects across all no-lockdown periods, underscoring the structural characteristics of Thailand's urban economy and transportation systems. As a large proportion of the population relies on in-person work environments and public transportation, particularly in major urban centers, these settings remained critical factors for virus transmission even in the absence of stringent government interventions.

## 2.2. Forecasting model

The detailed findings in section 2.1 indicate that the extent to which the mobility pattern derived from Google Community Mobility Reports (GCMR) and governmental policy interventions has a substantial impact on the incidence of confirmed COVID-19 cases in Thailand. These insights emphasize the critical role of mobility data in shaping effective public health responses during pandemics. These findings led us to proceed with our research question no. 2: "Can mobility data collected during a pandemic be utilized to model and forecast population movement patterns that support the implementation of appropriate government interventions to limit disease transmission?". In doing so, we pursued a certain study on related literature, and to the best of our knowledge, we found that Facebook Prophet is the latest approach that has been used to make similar forecasts. We also worked with the contemporary statistical approach, choosing ARIMA as it is a well-renowned method for time series data. We then decided to use one of the techniques of ensemble learning called XGBoost and modified it with feature engineering to make the parameters more suited to our dataset, and we finally named it "Feature Engineered XGBoost". These three approach are then used to predict mobility changes in Thailand

during the COVID-19 pandemic. All models are trained on the historical Google Community Mobility Report (GCMR) series and evaluated on their one-step-ahead accuracy using a hold-out test set. The models were implemented using Python 3.9. Forecasting was conducted using the following major libraries: Facebook Prophet (v1.1) for additive time series decomposition, pmdarima (v2.0.3) with the auto_arima() function for automated ARIMA modeling, and XGBoost (v1.7.6) using the XGBRegressor for gradient boosting regression. The following are detailed discussion of those three approaches using for forecasting.

**2.2.1. Facebook prophet.** Facebook Prophet is an additive time-series model designed for business data with multiple seasonalities and changepoints. Its form is:

$$y(t) = g(t) + s(t) + h(t) + \varepsilon_t \tag{1}$$

where $g(t)$ is a piecewise-linear (or logistic) trend component with automatically detected changepoints, $s(t)$ captures periodic effects (e.g., weekly and yearly seasonality), $h(t)$ models holidays or events, and $\varepsilon_t$ is the error term.

To optimize Prophet's forecasting performance in the Thai pandemic context, key hyperparameters were adjusted. Low changepoint and seasonality prior scales (0.05) reduced overfitting, while a high holiday prior scale (13.0) captured mobility shifts during major Thai holidays (e.g., Songkran, New Year). [26] Linear growth was applied to reflect steady trends, and multiple seasonalities along with Thai-specific holidays were incorporated to model recurring behavioral patterns.

In addition to the default seasonal components, we incorporated custom seasonalities specific to Thailand to capture the periodic behavioral changes associated with different waves of the COVID-19 pandemic and the implementation of lockdown measures. Specifically, four pandemic waves as shown in Table 1 and five distinct lockdown periods as shown in Table 2 were identified and modeled individually.

**2.2.2. ARIMA.** Autoregressive Integrated Moving Average (ARIMA) is a classical statistical model for time series forecasting. It combines three components: an autoregressive (AR) term that models the dependence on previous values, a differencing (I) component to ensure stationarity, and a moving average (MA) term to account for residual errors.

$$y_t = c + \varphi_1 y_{t-1} + \varphi_2 y_{t-2} + \ldots + \varphi_p y_{t-p} + \theta_1 \varepsilon_{t-1} + \theta_2 \varepsilon_{t-2} + \ldots + \theta_q \varepsilon_{t-q} + \varepsilon_t \tag{2}$$

where $y_t$ is the observed value at time $t$, $c$ is a constant term, $\varphi_i$ are the coefficients of the autoregressive (AR) terms, $\theta_i$ are the coefficients of the moving average (MA) terms, and $\varepsilon_t$ is the white noise error term at time $t$.

In our study, we employed the auto_arima() function from the pmdarima package, which automates the selection of optimal ARIMA hyperparameters based on the Akaike Information Criterion or AIC [27,28]. Each of the six Google Mobility indicators was modeled individually using ARIMA.

Despite ARIMA's interpretability and simplicity, it lacks the flexibility to incorporate exogenous variables (e.g., COVID-19 case rates, lockdown levels) in its basic form. Therefore, its performance was used as a benchmark against more complex models like Prophet and XGBoost. Nonetheless, ARIMA captured core temporal structures such as trend and seasonality effectively in the short term.

**2.2.3. XGBoost.** The proposed method Feature Engineered XGBoost is based on Extreme Gradient Boosting (XGBoost) with extended feature engineering. This modification allows the powerful ensemble learning method based on decision trees to work more precisely on the complex and essential features of the data. Unlike ARIMA or Prophet, XGBoost is not inherently designed for time series data but can effectively model complex relationships when augmented with engineered temporal and contextual features.

$$\hat{y}_i = \sum_{k=1}^{K} f_k(x_i), \quad f_k \in F \tag{3}$$

$$x_i = \left[ \underbrace{DOW_i, DOM_i, WN_i, M_i, Y_i, Wknd_i, Holiday_i}_{\text{Date–derived features}}, \underbrace{Avg\_all\_mobility_i}_{\text{Average mobility}}, \underbrace{StringencyIndex_i}_{\text{Policy stringency}}, \underbrace{LockdownLevel_i}_{\text{Custom–coded lockdown}} \right]$$

where $\hat{y}_i$ is the predicted value for instance $i$, $f_k$ is the k-th decision tree, $F$ is the space of all possible regression trees.

The input feature vector $x_i$ incorporates a comprehensive set of predictors designed to capture temporal patterns, policy interventions, and behavioral dynamics during the pandemic. Specifically, $x_i$ includes date-derived features—day of week ($DOW_i$), day of month ($DOM_i$), week number ($WN_i$), month ($M_i$), year ($Y_i$), weekend indicator ($Wknd_i$), and holiday indicator ($Holiday_i$)—alongside policy-related variables such as the Stringency Index ($StringencyIndex_i$), custom-coded lockdown levels ($LockdownLevel_i$), and average mobility ($Avg\_all\_mobility_i$). This rich feature set enables the model to learn nuanced relationships between calendar effects, public health measures, and human mobility trends. Hyperparameters were rigorously tuned for optimal performance, with the best configuration using a learning rate of 0.1 and a maximum tree depth of 5 [29].

## 2.3. Evaluation metrics

To evaluate the accuracy of the forecasting models, three widely accepted performance metrics were employed: Mean Absolute Error (MAE), Root Mean Square Error (RMSE), and Mean Absolute Percentage Error (MAPE) [30]. Beyond the use of error-based metrics, model interpretability and explanatory insights were obtained through the analysis of feature importance scores derived from tree-based machine learning models. Specifically, importance was assessed based on three criteria [31]: weight (the number of times a feature is used to split the data), gain (the improvement in accuracy brought by a feature), and cover (the number of observations affected by a feature). These scores were used to identify the most influential variables driving changes in mobility patterns. Additionally, the Granger Causality Test was conducted to assess whether certain explanatory variables could statistically forecast future values of mobility, thus revealing potential temporal causal relationships.

We use heatmap as shown in Fig 3 to present Pearson correlation coefficients between different mobility categories (Retail & Recreation, Grocery & Pharmacy, Parks, Transit Stations, Workplaces, and Residential), the government policy stringency index, and the number of new confirmed COVID-19 cases. Strong positive correlations are observed among mobility categories such as Retail & Recreation and Grocery & Pharmacy, while negative correlations are observed between the Stringency Index and mobility indicators (e.g., Transit Stations: –0.77, Retail & Recreation: –0.68). Notably, Residential mobility is negatively correlated with most other categories and positively correlated with new COVID-19 cases ($r = 0.25$), suggesting increased residential activity during peak outbreaks. However, overall correlations between new COVID-19 cases and mobility indicators are relatively weak, indicating a complex and potentially lagged relationship.

## 3. Results and discussion

The dataset was divided into training and testing sets using a 90:10 ratio, where 90% of the data was used for model training and the remaining 10% for testing. This split was selected to provide the models with sufficient historical data to learn underlying mobility patterns, especially during the dynamic and rapidly evolving context of the COVID-19 pandemic. Given the limited data availability for certain phases of the outbreak and the high variability in human behavior due to sudden policy changes, a 90:10 split helped preserve enough sequential data for training while maintaining a representative and temporally relevant subset for performance evaluation. This approach is consistent with best practices in time series forecasting, especially in situations where the data exhibits non-stationarity and is subject to sudden changes caused by external events [24,25].

Comparison of actual and forecasted mobility data across six categories using FB-PROPhet, ARIMA, and XGBoost. Green lines represent training data, Blue lines represent test data, and Red lines represent model forecasts, as depicted in Figs 4–6.

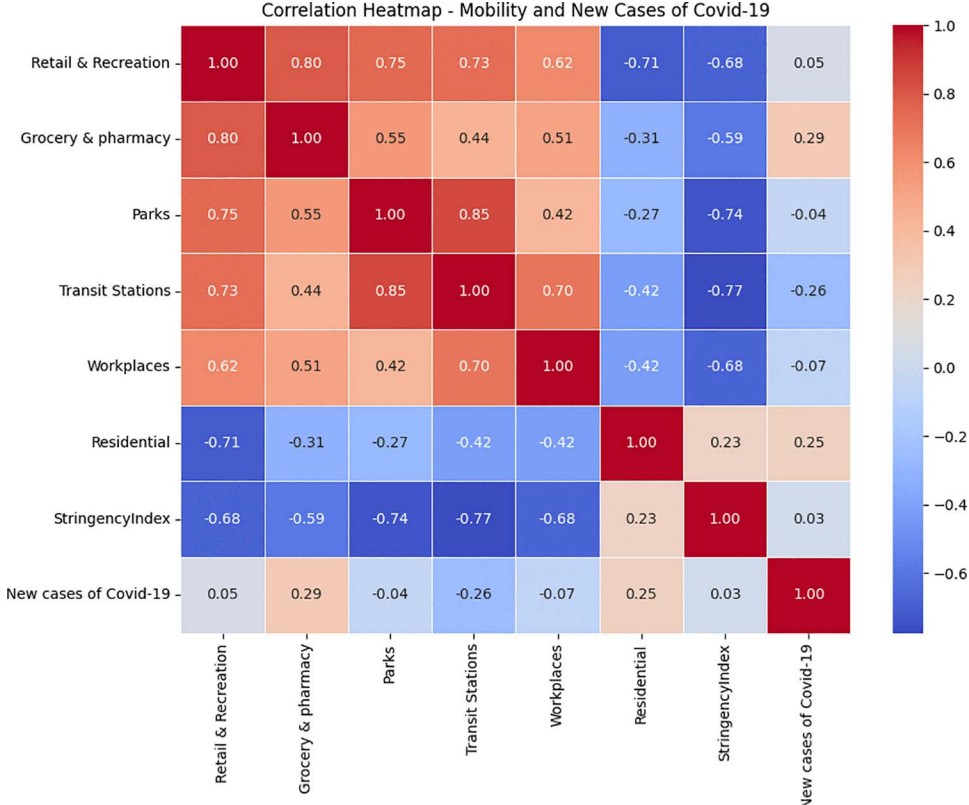

**Fig 3. Correlation heatmap showing relationships among mobility trends, policy stringency, and new COVID-19 case counts.**

Firstly, Fig 4 shows the performance of the ARIMA forecasts across the same six mobility categories. ARIMA tends to produce smoother and more conservative predictions, making it less responsive to the short-term fluctuations seen in the test data. Among the categories, residential mobility is where ARIMA performs relatively well, aligning most closely with the overall level of the actual values, despite missing subtle variations. Retail and recreation, as well as workplaces, display moderate accuracy, capturing general average levels but failing to represent the underlying dynamic patterns. In contrast, the grocery and pharmacy forecasts reveal a noticeable gap, with ARIMA unable to keep up with the higher variability reflected in the observations. The poorest fit is found in the parks and transit stations categories, where predicted values diverge sharply from the actual test data. These results highlight ARIMA's difficulty in modeling categories characterized by higher volatility and rapid changes.

Next, Fig 5 presents six mobility-category forecasts alongside their observed values. Overall, the results show that the forecasting quality varies noticeably across categories. The grocery and pharmacy forecast stands out as the most accurate, closely matching the timing and size of the peaks and dips in the test period. Retail and recreation also perform fairly well, generally following the direction and scale of the observed changes, although minor timing shifts and small amplitude errors are still present. Forecasts for workplaces and residential mobility are reasonably acceptable, capturing the broader level shifts but showing lag and some misestimation of peak levels, which makes them less reliable for short-term decision-making. The weakest outcomes appear in transit stations and parks, where the forecasts diverge substantially from the actual values. Large errors in both amplitude and timing suggest that the model has difficulty dealing with the irregular and volatile dynamics characteristic of these categories.

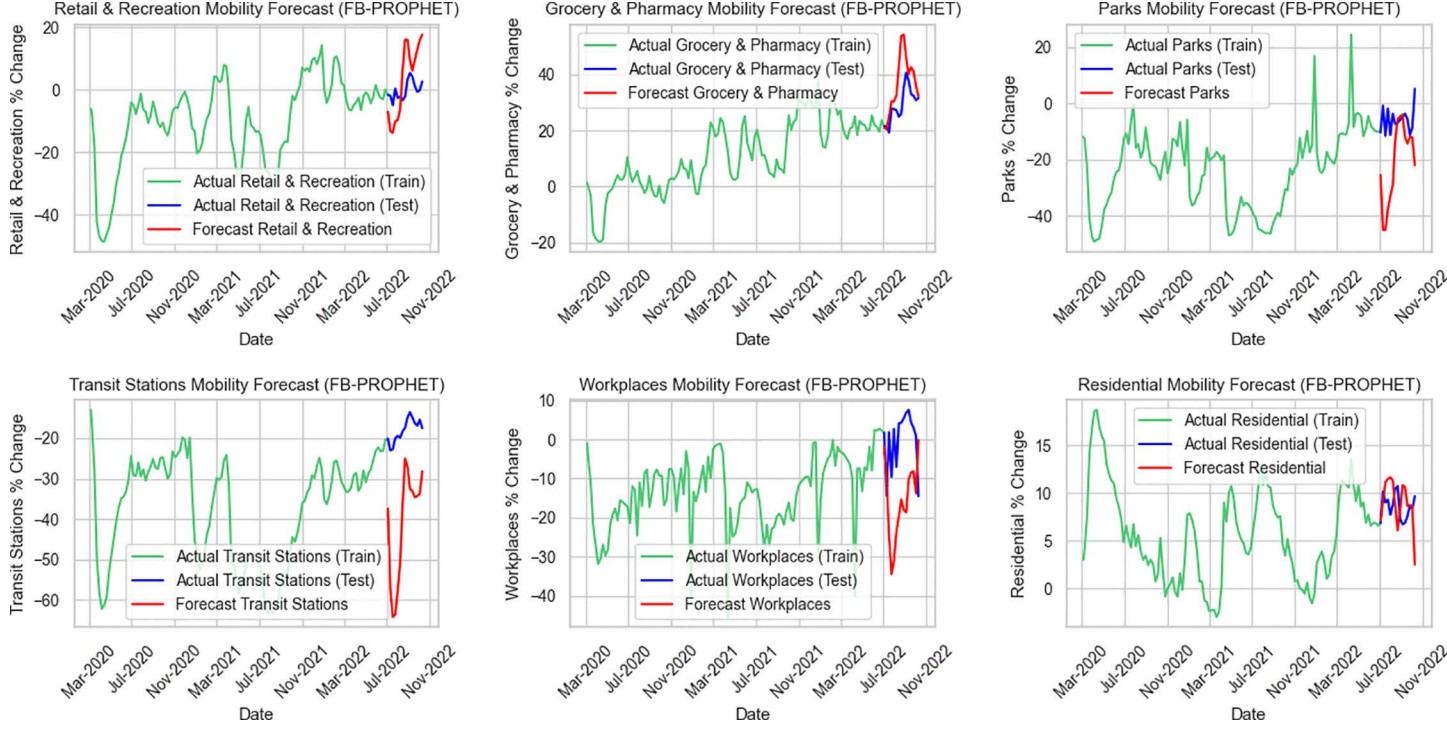

**Fig 4. Forecasted mobility changes and actual data in Thailand using ARIMA model.**

Continue to Fig 6, which illustrates that the Feature Engineered XGBoost model delivers consistently better predictive performance than both ARIMA and Facebook Prophet across all six mobility categories. Its forecasts follow the actual test data more closely, capturing both overall levels and short-term fluctuations with less bias and fewer deviations. The strongest results appear in grocery and pharmacy, as well as retail and recreation, where forecast values align well with the observed ranges. Workplaces and transit stations also benefit from Feature Engineered XGBoost's improved accuracy, avoiding the large under- or over-estimations that occur in ARIMA and Prophet forecasts. Even in more volatile categories, such as parks and residential mobility, the model still maintains better directional consistency and remains closer to the observed patterns. Overall, the results show that Feature Engineered XGBoost provides the most stable and reliable forecasts among the three models, offering substantially improved accuracy for mobility prediction during the test period.

In addition to using the line graph to demonstrate the forecasting accuracy of the three models—ARIMA, FB-PROPHET, and Feature Engineered XGBoost—we also compared their performance across six mobility-related location categories using three standard error metrics: Mean Absolute Error (MAE), Root Mean Square Error (RMSE), and Mean Absolute Percentage Error (MAPE). The results are summarized in Table 6.

As shown in Table 6, ARIMA is used as the baseline model to compare with FB-PROPHET and XGBoost. Across all categories, Feature Engineered XGBoost consistently outperforms both ARIMA and FB-PROPHET, achieving significant error reductions in MAE, RMSE, and MAPE. For instance, in Retail and Recreation, Feature Engineered XGBoost reduces MAE by 40.6% compared to ARIMA, while FB-PROPHET shows negative performance (worse than ARIMA). Similarly, in Parks and Transport stations, Feature Engineered XGBoost achieves over 60% error reduction in MAE and RMSE, whereas FB-PROPHET performs substantially worse than the baseline. Overall, ARIMA serves as a stable baseline, but Feature Engineered XGBoost appears as the most accurate forecasting model across all location types.

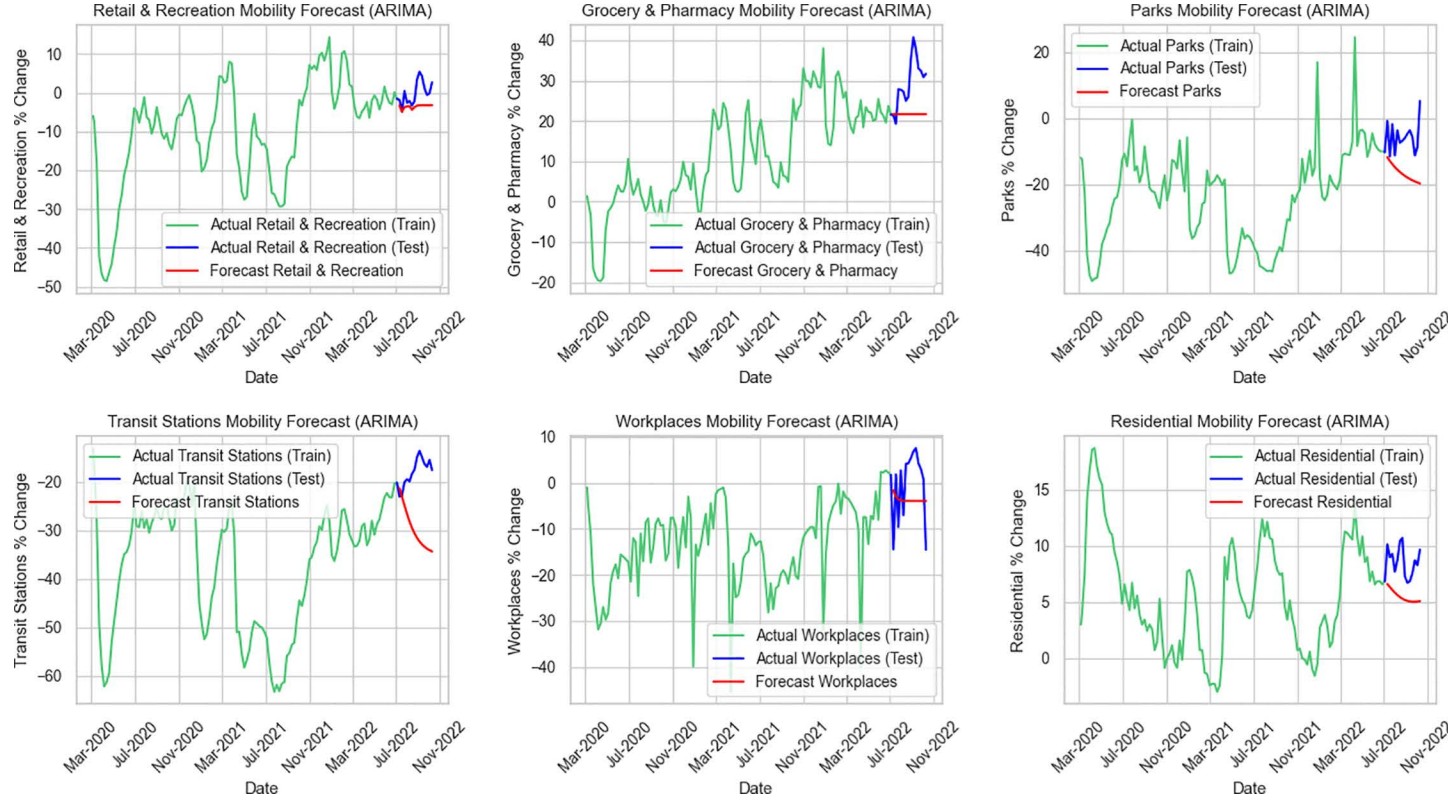

**Fig 5. Forecasted mobility changes and actual data in Thailand using Facebook Prophet model.**

These findings highlight the strength of the machine learning model used in this study, "Feature Engineered XGBoost," in handling real-world, high-variance time series data, especially when traditional statistical methods like ARIMA or trend-based models like FB-PROPHET are limited in capturing abrupt shifts due to external events such as policy changes or outbreak surges.

In order to ensure the highest level of predictive accuracy and model stability, this study employed a dual-validation framework. As discussed earlier, the dataset was evaluated using a standard 90:10 training-to-testing ratio to establish baseline performance. Subsequently, a rigorous Rolling-Origin Evaluation (ROE) strategy was applied to all three forecasting models: ARIMA, Prophet, and Feature Engineered XGBoost (the proposed method). The ROE approach, as illustrated in Fig 7, utilizes an expanding window technique where the forecast origin shifts chronologically across four distinct pandemic phases in Thailand: the Wuhan (Wave 1), Alpha-Beta (Wave 2), Delta (Wave 3), and Omicron (Wave 4) waves.

Fig 7 compares MAE values across six mobility categories, namely Retail and Recreation, Grocery and Pharmacy, Parks, Transit Stations, Workplaces, and Residential, over four COVID-19 waves in Thailand using ARIMA, FB-PROPHET and XGBoost based on Rolling-Origin Evaluation Approach. Overall, the Feature Engineered XGBoost model consistently achieves the lowest prediction errors across most mobility categories and pandemic phases, particularly for mobility types that are highly sensitive to policy interventions such as Transit Stations, Workplaces, and Residential areas. During the first wave (Wuhan), prediction errors remain relatively moderate across all models, reflecting more uniform mobility restrictions. More pronounced performance differences emerge in subsequent waves as mobility behavior becomes increasingly heterogeneous. In the Alpha-Beta and Delta waves, MAE values increase notably, especially for Parks and Workplaces, indicating heightened volatility and reduced predictability during periods of rapid and stringent policy changes. In contrast,

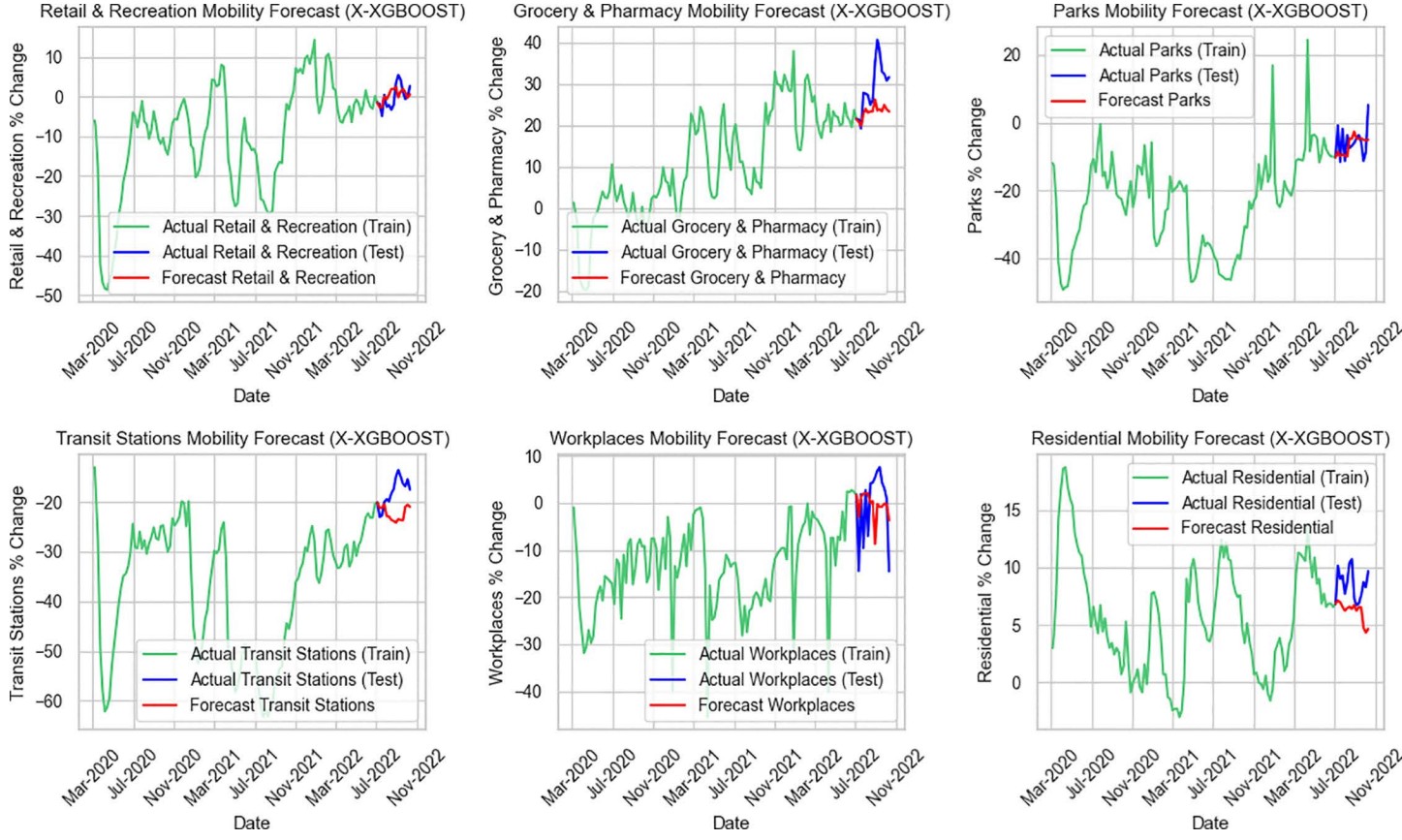

**Fig 6. Forecasted mobility changes and actual data in Thailand using Feature Engineered XGBoost model.**

the Omicron wave exhibits lower MAE values in several categories, suggesting partial behavioral adaptation and stabilization despite ongoing policy adjustments. ARIMA performs reasonably well in mobility categories with smoother trends but shows reduced accuracy during highly variable periods, while Prophet records the highest MAE in most categories, particularly for Parks and Workplaces.

Apart from the process of selecting features to improve model performance (feature engineering) during preprocessing, we also perform feature importance analysis to see which features influenced the model the most. As shown in Table 7, the temporal variables continue to play a dominant role in mobility prediction, with "day" and "week" emerging as consistently influential across all six categories. Although "day" remains one of the strongest predictors, the results now highlight that "avg_all_mobility" carries exceptionally high weight and gain across categories such as Retail & Recreation (30.25% weight, 55.50% gain), Parks (27.87% weight, 58.45% gain), Transport Stations (25.87% weight, 49.67% gain), and Workplaces (25.21% weight, 32.88% gain). This indicates that overall mobility tendencies strongly shape category-specific movements. In addition, "year" shows notably high gain in some categories—for example, 89.19% in Grocery Stores & Pharmacies—suggesting that broader annual patterns or pandemic phase shifts also influence mobility dynamics.

Regarding pandemic-related variables, the Stringency Index remains a consistently important predictor, with its importance reflected strongly in Workplaces, Residential Areas, Transport Stations, and Retail & Recreation—highlighting the impact of nationwide policy measures on mobility behavior. Meanwhile, lockdown_level shows moderate importance but contributes significantly to certain categories such as Residential and Workplaces, where regulatory changes directly

**Table 6. Forecast accuracy of mobility trends using ARIMA, FB-PROPHET and Feature Engineered XGBoost, across location categories.**

| Category | Model | MAE | RMSE | MAPE |
|---|---|---|---|---|
| Retail and Recreation | ARIMA (Baseline) | 3.55 | 4.33 | 3.44 |
| | FB-PROPHET<br>Error reduction | 9.79<br>(−175.8%) | 10.82<br>(−149.9%) | 13.75<br>(−299.7%) |
| | XGBoost (Proposed)<br>Error reduction | 2.11<br>(40.6%) | 2.70<br>(37.6%) | 0.95<br>(72.4%) |
| Grocery stores and pharmacies | ARIMA (Baseline) | 7.97 | 9.70 | 0.24 |
| | FB-PROPHET<br>Error reduction | 7.64<br>(4.1%) | 10.87<br>(−12.1%) | 0.26<br>(−8.3%) |
| | XGBoost (Proposed)<br>Error reduction | 6.06<br>(24.0%) | 7.76<br>(20.0%) | 0.18<br>(25.0%) |
| Parks | ARIMA (Baseline) | 10.97 | 12.29 | 3.08 |
| | FB-PROPHET<br>Error reduction | 15.54<br>(−41.7%) | 21.05<br>(−71.3%) | 6.31<br>(−104.9%) |
| | XGBoost (Proposed)<br>Error reduction | 3.67<br>(66.6%) | 4.77<br>(61.2%) | 1.32<br>(57.1%) |
| Transport stations | ARIMA (Baseline) | 12.01 | 13.73 | 0.73 |
| | FB-PROPHET<br>Error reduction | 23.00<br>(−91.5%) | 25.56<br>(−86.2%) | 1.24<br>(−70.0%) |
| | XGBoost (Proposed)<br>Error reduction | 4.67<br>(61.1%) | 5.58<br>(59.4%) | 0.29<br>(60.3%) |
| Workplaces | ARIMA (Baseline) | 7.60 | 8.07 | 1.88 |
| | FB-PROPHET<br>Error reduction | 18.21<br>(−139.6%) | 19.96<br>(−147.3%) | 5.32<br>(−183.0%) |
| | XGBoost (Proposed)<br>Error reduction | 6.07<br>(20.1%) | 7.153<br>(11.4%) | 0.99<br>(47.3%) |
| Residential areas | ARIMA (Baseline) | 3.07 | 3.35 | 0.34 |
| | FB-PROPHET<br>Error reduction | 2.39<br>(22.2%) | 3.07<br>(8.4%) | 0.28<br>(17.6%) |
| | XGBoost (Proposed)<br>Error reduction | 2.33<br>(24.1%) | 2.82<br>(15.8%) | 0.25<br>(26.5%) |

Note: Error reduction (%) = [(ARIMA_Error − Model_Error)/ ARIMA_Error] * 100.

affected daily routines. The importance of is_holiday also rises in specific categories like Residential Areas and Workplaces, aligning with expected seasonal and cultural mobility shifts.

In addition to the tabular form of information about important features, we have also conducted an analysis using a bar graph, as illustrated in Fig 8, to depict the influencing level of each important feature. This graph shows the frequency of feature usage in trees (weight), the contribution to loss reduction (gain), and the relative number of samples affected (cover), as shown below.

Fig 8 illustrates the feature importance analysis of the Feature Engineered XGBoost model for population mobility forecasting across all mobility categories, including Retail and Recreation, Grocery and Pharmacy, Parks, Transit Stations, Workplaces, and Residential. The analysis is evaluated using three complementary importance metrics: Weight, Gain, and Cover, which respectively represent the frequency of feature usage in tree construction, the contribution of each feature to loss reduction, and the proportion of samples affected by feature-based splits.

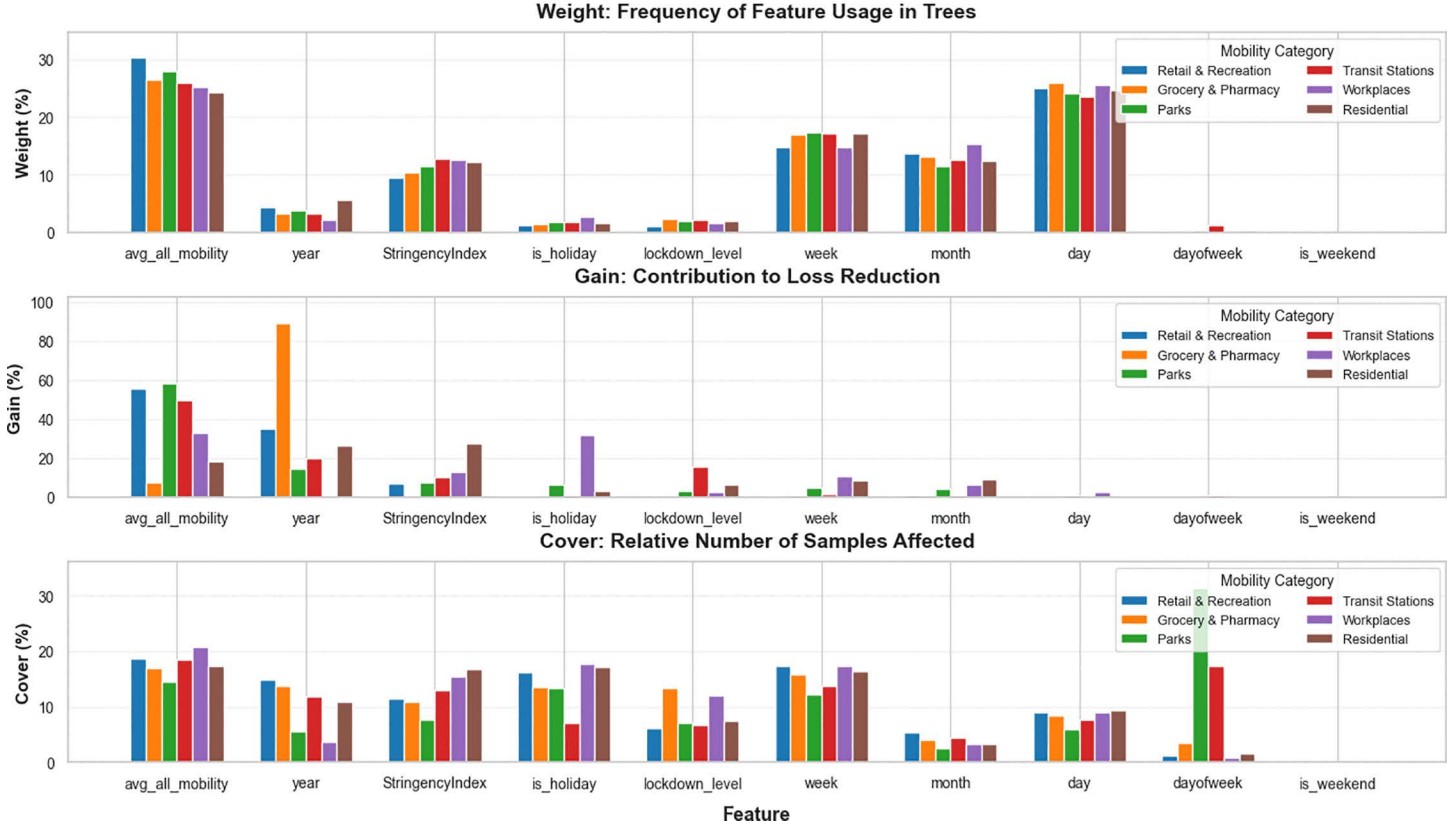

**Fig 7. Comparison of MAE across 4 COVID-19-Waves using ARIMA, FB-PROPHET and XGBoost based on Rolling-Origin Evaluation Approach.**

In the Weight dimension (top panel), which reflects how frequently features are selected during tree construction, avg_all_mobility and temporal features such as day, week, and month consistently exhibit the highest values across all mobility categories. This indicates that historical mobility patterns and temporal structures form the core foundation of the forecasting process. The StringencyIndex plays a secondary but notable role, particularly in policy-sensitive categories such as Workplaces and Transit Stations. In contrast, policy-specific variables such as lockdown_level and is_holiday are used less frequently, suggesting that they function primarily as complementary rather than dominant predictors.

The Gain dimension (middle panel), which measures each feature's contribution to reducing prediction error, reveals clear structural differences across mobility categories. The feature avg_all_mobility provides substantial gain for Retail and Recreation, Parks, and Transit Stations, while year shows a pronounced influence in Grocery and Pharmacy and Residential mobility, reflecting longer-term structural changes across pandemic periods. Additionally, is_holiday and lockdown_level demonstrate relatively high gain in specific categories such as Workplaces and Transit Stations, indicating strong but localized effects associated with policy interventions or special periods.

In the Cover dimension (bottom panel), which represents the relative proportion of samples influenced by feature splits, temporal variables including avg_all_mobility, year, week, and day consistently affect a large share of observations across all mobility categories. The StringencyIndex and lockdown_level exhibit moderate coverage, particularly in work and travel

**Table 7. Feature importance for mobility prediction across six location categories (Weight, Gain, and Cover).**

| Category | Feature | Weight (%) | Gain (%) | Cover (%) | Category | Feature | Weight (%) | Gain (%) | Cover (%) |
|---|---|---|---|---|---|---|---|---|---|
| Retail and Recreation | dayofweek | 0.02 | 0.00 | 1.17 | Transport stations | dayofweek | 1.13 | 0.88 | 17.32 |
| | month | 13.58 | 1.03 | 5.2 | | month | 12.53 | 0.94 | 4.32 |
| | day | 25.00 | 0.10 | 8.92 | | day | 23.49 | 0.73 | 7.66 |
| | week | 14.82 | 0.73 | 17.34 | | week | 17.05 | 1.84 | 13.71 |
| | year | 4.39 | 35.20 | 14.78 | | year | 3.1 | 19.69 | 11.82 |
| | is_holiday | 1.29 | 0.30 | 16.25 | | is_holiday | 1.82 | 0.83 | 7.03 |
| | lockdown_level | 1.12 | 0.20 | 6.02 | | lockdown_level | 2.14 | 15.35 | 6.72 |
| | stringency_index | 9.51 | 6.93 | 11.52 | | stringency_index | 12.78 | 10.05 | 13.02 |
| | avg_all_mobility | 30.25 | 55.50 | 18.70 | | avg_all_mobility | 25.87 | 49.67 | 18.41 |
| Grocery stores and pharmacies | dayofweek | 0.06 | 0.07 | 3.46 | Workplaces | dayofweek | 0.20 | 0.00 | 0.84 |
| | month | 13.06 | 1.31 | 3.98 | | month | 15.24 | 6.19 | 3.30 |
| | day | 25.99 | 0.17 | 8.36 | | day | 25.5 | 2.54 | 9.02 |
| | week | 17.03 | 0.78 | 15.81 | | week | 14.83 | 10.54 | 17.27 |
| | year | 3.19 | 89.19 | 13.77 | | year | 2.11 | 0.54 | 3.59 |
| | is_holiday | 1.37 | 0.16 | 13.58 | | is_holiday | 2.75 | 32.01 | 17.73 |
| | lockdown_level | 2.36 | 0.28 | 13.29 | | lockdown_level | 1.59 | 2.42 | 12.05 |
| | stringency_index | 10.45 | 0.55 | 10.86 | | stringency_index | 12.52 | 12.88 | 15.40 |
| | avg_all_mobility | 26.49 | 7.48 | 16.88 | | avg_all_mobility | 25.21 | 32.88 | 20.81 |
| Parks | dayofweek | 0.35 | 0.00 | 31.41 | Residential areas | dayofweek | 0.37 | 0.30 | 1.53 |
| | month | 11.50 | 4.04 | 2.53 | | month | 12.33 | 9.25 | 3.22 |
| | day | 24.10 | 0.82 | 5.95 | | day | 24.57 | 0.22 | 9.30 |
| | week | 17.24 | 5.08 | 12.11 | | week | 17.10 | 8.39 | 16.45 |
| | year | 3.75 | 14.49 | 5.60 | | year | 5.54 | 26.28 | 10.88 |
| | is_holiday | 1.82 | 6.27 | 13.29 | | is_holiday | 1.62 | 3.00 | 17.15 |
| | lockdown_level | 1.99 | 3.33 | 7.00 | | lockdown_level | 1.95 | 6.52 | 7.46 |
| | stringency_index | 11.38 | 7.53 | 7.57 | | stringency_index | 12.22 | 27.64 | 16.69 |
| | avg_all_mobility | 27.87 | 58.45 | 14.55 | | avg_all_mobility | 24.31 | 18.41 | 17.33 |

related categories. Notably, day_of_week shows high coverage in Parks and Transit Stations, highlighting systematic differences in mobility behavior between weekdays and weekends.

## 4. Conclusions

This study introduced a comprehensive approach for forecasting mobility trends to support crisis-responsive mobility management. Using Google Community Mobility Reports across six key categories, the proposed Feature Engineered XGBoost model demonstrated superior performance compared with other baseline models. The expanded feature importance analysis clearly shows that temporal factors (day, week, and year) remain the strongest drivers of mobility fluctuations, while aggregated mobility measures (avg_all_mobility) serve as powerful indicators of category-level movement patterns. Pandemic-related variables—particularly the Stringency Index, lockdown restrictions, and holiday effects—also significantly shaped mobility variations, reflecting how policy interventions and social cycles influenced public behavior throughout the pandemic.

This study highlights the importance of selecting modeling techniques that align with the specific context and characteristics of the data, especially in the dynamic and complex environment of a public health crisis. We hope that our findings will serve as a valuable foundation for policymakers, researchers, and public health professionals aiming to interpret and

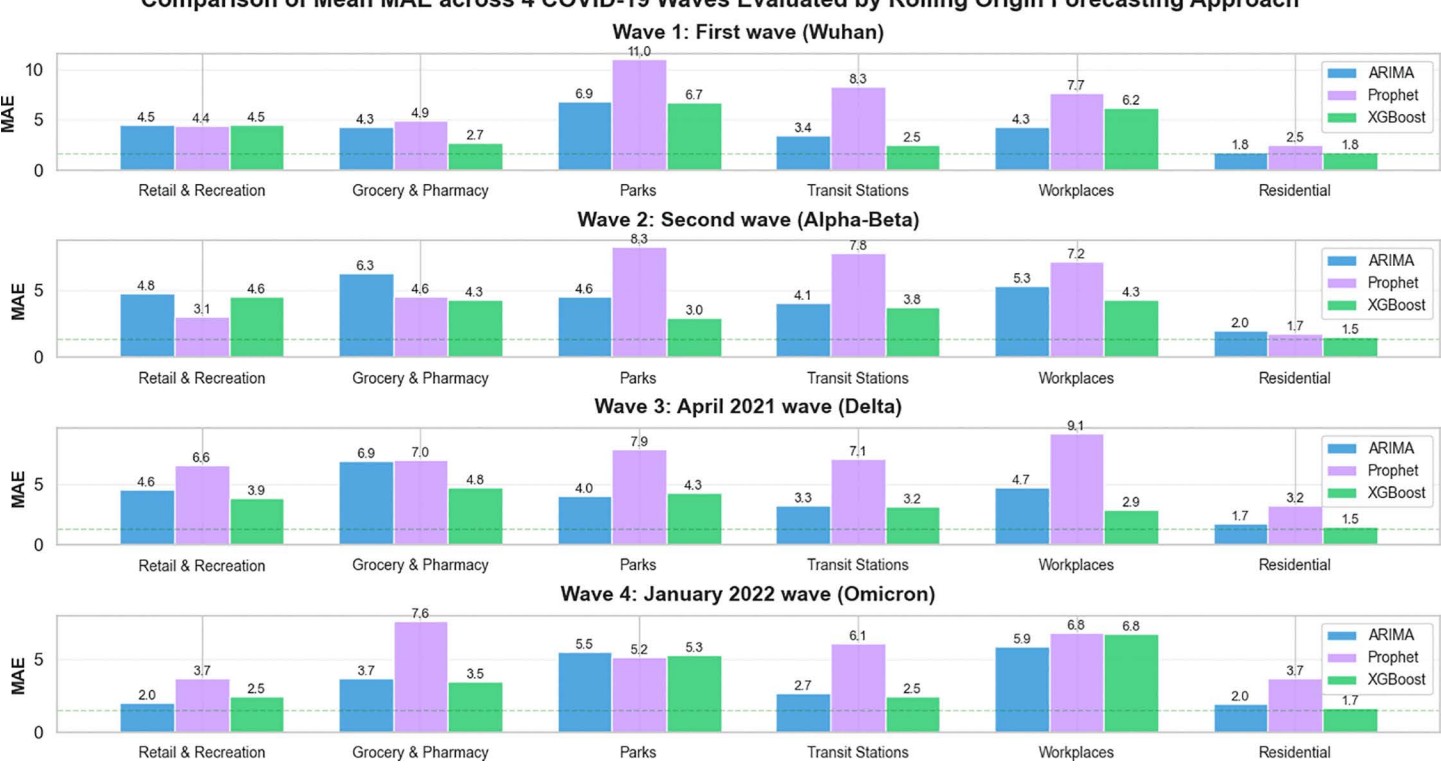

**Fig 8. Feature importance analysis of the Feature Engineered XGBoost Approach.**

respond to mobility trends during such emergencies. Additionally, we encourage future research to expand on this work by incorporating mobility data from multiple countries, including a broader range of behavioral and socio-economic variables, and exploring advanced methods such as deep learning to capture more nuanced and complex patterns in mobility dynamics.

## Supporting information

**S1 Text. Python Code for ARIMA approach.**
(TXT)

**S2 Text. Python Code for FB-Prophet approach.**
(TXT)

**S3 Text. Python Code for XGBoost approach.**
(TXT)

**S4 Text. Python Code for Rolling-origin approach.**
(TXT)

**S1 Dataset. GMR.** Sample dataset of Google Mobility Reports.
(CSV)

**S2 Dataset. JHU.** Sample dataset of Johns Hopkins University Center for Systems Science and Engineering (JHU-CSSE).
(CSV)

**S3 Dataset. Oxford.** Sample dataset of the Blavatnik School of Government, University of Oxford.
(XLSX)

## Author contributions

**Conceptualization:** Aritath Siraphatwongkorn, Thanin Methiyothin, Kittisak Onuean, Krisana Chinnasarn, Athita Onuean, Insung Ahn, Suwanna Rasmequan.

**Data curation:** Aritath Siraphatwongkorn.

**Formal analysis:** Aritath Siraphatwongkorn, Thanin Methiyothin, Kittisak Onuean, Krisana Chinnasarn, Athita Onuean, Insung Ahn, Suwanna Rasmequan.

**Funding acquisition:** Insung Ahn.

**Investigation:** Aritath Siraphatwongkorn.

**Methodology:** Aritath Siraphatwongkorn, Thanin Methiyothin, Kittisak Onuean, Krisana Chinnasarn, Athita Onuean, Insung Ahn, Suwanna Rasmequan.

**Project administration:** Insung Ahn.

**Supervision:** Insung Ahn, Suwanna Rasmequan.

**Validation:** Thanin Methiyothin, Kittisak Onuean, Krisana Chinnasarn, Athita Onuean, Suwanna Rasmequan.

**Writing – original draft:** Aritath Siraphatwongkorn.

**Writing – review & editing:** Suwanna Rasmequan.

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
