## [Decision Letter · Decision Letter 0]

28 Jan 2026

Dear Dr. Rasmequan,

Thank you for submitting your manuscript to PLOS ONE. After careful consideration, we feel that it has merit but does not fully meet PLOS ONE’s publication criteria as it currently stands. Therefore, we invite you to submit a revised version of the manuscript that addresses the points raised during the review process.

We look forward to receiving your revised manuscript.

Kind regards,

Matthew Chin Heng Chua

Academic Editor

PLOS One

Journal Requirements:

2. In your Methods section, please include additional information about your dataset and ensure that you have included a statement specifying whether the collection and analysis method complied with the terms and conditions for the source of the data.

3. Please note that PLOS One has specific guidelines on code sharing for submissions in which author-generated code underpins the findings in the manuscript. In these cases, we expect all author-generated code to be made available without restrictions upon publication of the work. Please review our guidelines at https://journals.plos.org/plosone/s/materials-and-software-sharing#loc-sharing-code and ensure that your code is shared in a way that follows best practice and facilitates reproducibility and reuse

“This research was supported by the Government-wide R&D to Advance Infectious Disease Prevention and Control, Republic of Korea (grant number: RS-2023-KH140419).

This research was also co-funding by Faculty of Informatics, Burapha University, Thailand.”

5. Please include captions for your Supporting Information files at the end of your manuscript, and update any in-text citations to match accordingly. Please see our Supporting Information guidelines for more information: http://journals.plos.org/plosone/s/supporting-information ..

Reviewers' comments:

Reviewer's Responses to Questions

**Comments to the Author**

1. Is the manuscript technically sound, and do the data support the conclusions?

Reviewer #1: Partly

2. Has the statistical analysis been performed appropriately and rigorously?

Reviewer #1: No

3. Have the authors made all data underlying the findings in their manuscript fully available?

Reviewer #1: Yes

4. Is the manuscript presented in an intelligible fashion and written in standard English?

Reviewer #1: Yes

Reviewer #1: The methodology is well-structured and follows a logical progression from data acquisition to model evaluation.

The use of three distinct, high-quality datasets—the Google Community Mobility Reports, the Oxford Stringency Index, and JHU-CSSE epidemiological data—provides a comprehensive view of Thailand's pandemic dynamics. Furthermore, integrating these via a "Date" primary key ensures temporal consistency.

Regarding the model validation, the paper employs a 90:10 train-test split. While the authors justify this as necessary to capture mobility patterns during rapid policy changes, a fixed split in time-series data is often problematic. I recommend implementing time-series cross-validation (such as a rolling origin approach) to demonstrate model stability across different pandemic waves.

Comparing a traditional statistical model (ARIMA), a trend-based model (Prophet), and a machine learning model (XGBoost) provides a robust benchmark. However, the nomenclature "X-XGBoost" requires clarification. In machine learning literature, "Extended" often implies an algorithmic modification. The authors should specify whether "X-XGBoost" involves an architectural change or simply refers to a standard XGBRegressor with a specific feature set. If the model relies solely on feature engineering, renaming it "Feature-Engineered XGBoost" would prevent confusion regarding potential algorithmic extensions.

While the methods effectively tie to the results, some gaps remain. For instance, although "feature importance" (weight and gain) is mentioned in the methods, these results are only briefly summarized. A detailed table or plot showing which features (e.g., Stringency Index vs. Day of Week) were most influential for each of the six mobility categories would better justify the "X-XGBoost" approach.

Finally, to enhance transparency and reproducibility of the machine learning implementation, I strongly recommend that the authors provide the full code or scripts used for data preprocessing, model training, and evaluation. Making these scripts available—either as supplementary material or via a public repository (e.g., GitHub)—would allow other researchers to replicate the results and verify the methodology

**Do you want your identity to be public for this peer review?** For information about this choice, including consent withdrawal, please see our For information about this choice, including consent withdrawal, please see our Privacy Policy .

Reviewer #1: No

---

## [Author Response · Author response to Decision Letter 1]

10 Feb 2026

Reviewers' comments

1. The methodology is well-structured and follows a logical progression from data acquisition to model evaluation.

The use of three distinct, high-quality datasets—the Google Community Mobility Reports, the Oxford Stringency Index, and JHU-CSSE epidemiological data—provides a comprehensive view of Thailand's pandemic dynamics. Furthermore, integrating these via a "Date" primary key ensures temporal consistency.

Regarding the model validation, the paper employs a 90:10 train-test split. While the authors justify this as necessary to capture mobility patterns during rapid policy changes, a fixed split in time-series data is often problematic. I recommend implementing time-series cross-validation (such as a rolling origin approach) to demonstrate model stability across different pandemic waves.

Response :

Thank you for your comment. For this comment, we have added a graph depicting the analysis of MAE of the validation results of each forecasting approach using a Rolling-Origin Evaluation (ROE) method in addition to a 90:10 train-test split. We have categorized the validation results into four event waves, as shown below. We hope this will help demonstrate the stability of our proposed method across different pandemic waves.

In order to ensure the highest level of predictive accuracy and model stability, this study employed a dual-validation framework. As discussed earlier, the dataset was evaluated using a standard 90:10 training-to-testing ratio to establish baseline performance. Subsequently, a rigorous Rolling-Origin Evaluation (ROE) strategy was applied to all three forecasting models: ARIMA, Prophet, and Feature Engineered XGBoost (the proposed method). The ROE approach, as illustrated in Figure 7, utilizes an expanding window technique where the forecast origin shifts chronologically across four distinct pandemic phases in Thailand: the Wuhan (Wave 1), Alpha-Beta (Wave 2), Delta (Wave 3), and Omicron (Wave 4) waves.

Figure 8. Comparison of MAE across 4 COVID-19-Waves using ARIMA, FB-PROPHET and XGBoost based on Rolling-Origin Evaluation Approach

Figure 8 compares MAE values across six mobility categories, namely Retail and Recreation, Grocery and Pharmacy, Parks, Transit Stations, Workplaces, and Residential, over four COVID-19 waves in Thailand using ARIMA, FB-PROPHET and Xgboost based on Rolling-Origin Evaluation Approach. Overall, the Feature Engineered XGBoost model consistently achieves the lowest prediction errors across most mobility categories and pandemic phases, particularly for mobility types that are highly sensitive to policy interventions such as Transit Stations, Workplaces, and Residential areas. During the first wave (Wuhan), prediction errors remain relatively moderate across all models, reflecting more uniform mobility restrictions. More pronounced performance differences emerge in subsequent waves as mobility behavior becomes increasingly heterogeneous. In the Alpha-Beta and Delta waves, MAE values increase notably, especially for Parks and Workplaces, indicating heightened volatility and reduced predictability during periods of rapid and stringent policy changes. In contrast, the Omicron wave exhibits lower MAE values in several categories, suggesting partial behavioral adaptation and stabilization despite ongoing policy adjustments. ARIMA performs reasonably well in mobility categories with smoother trends but shows reduced accuracy during highly variable periods, while Prophet records the highest MAE in most categories, particularly for Parks and Workplaces.

2. Comparing a traditional statistical model (ARIMA), a trend-based model (Prophet), and a machine learning model (XGBoost) provides a robust benchmark. However, the nomenclature "X-XGBoost" requires clarification. In machine learning literature, "Extended" often implies an algorithmic modification. The authors should specify whether "X-XGBoost" involves an architectural change or simply refers to a standard XGBRegressor with a specific feature set. If the model relies solely on feature engineering, renaming it "Feature-Engineered XGBoost" would prevent confusion regarding potential algorithmic extensions.

Response :

Thank you for your comment. We have changed X-XGBoost to Feature Engineered XGBoost and also made change to our title too.

3. While the methods effectively tie to the results, some gaps remain. For instance, although "feature importance" (weight and gain) is mentioned in the methods, these results are only briefly summarized. A detailed table or plot showing which features (e.g., Stringency Index vs. Day of Week) were most influential for each of the six mobility categories would better justify the "X-XGBoost" approach.

Response :

Thank you for your comment. For this comment, we have added additional analysis using bar-graph to depict the influencing level to the XGBoost model of each feature important and also added the explanation as shown below. Hope, this could help reader to see the significant of each feature better.

In addition to the tabular form of information about important features, we have also conducted an analysis using a bar graph, as illustrated in Figure 9, to depict the influencing level of each important feature. This graph shows the frequency of feature usage in trees (weight), the contribution to loss reduction (gain), and the relative number of samples affected (cover), as shown below.

Figure 9. Feature importance analysis of the Feature Engineered XGBoost Approach

Figure 9 illustrates the feature importance analysis of the Feature Engineered XGBoost model for population mobility forecasting across all mobility categories, including Retail and Recreation, Grocery and Pharmacy, Parks, Transit Stations, Workplaces, and Residential. The analysis is evaluated using three complementary importance metrics: Weight, Gain, and Cover, which respectively represent the frequency of feature usage in tree construction, the contribution of each feature to loss reduction, and the proportion of samples affected by feature-based splits.

In the Weight dimension (top panel), which reflects how frequently features are selected during tree construction, avg_all_mobility and temporal features such as day, week, and month consistently exhibit the highest values across all mobility categories. This indicates that historical mobility patterns and temporal structures form the core foundation of the forecasting process. The StringencyIndex plays a secondary but notable role, particularly in policy-sensitive categories such as Workplaces and Transit Stations. In contrast, policy-specific variables such as lockdown_level and is_holiday are used less frequently, suggesting that they function primarily as complementary rather than dominant predictors.

The Gain dimension (middle panel), which measures each feature’s contribution to reducing prediction error, reveals clear structural differences across mobility categories. The feature avg_all_mobility provides substantial gain for Retail and Recreation, Parks, and Transit Stations, while year shows a pronounced influence in Grocery and Pharmacy and Residential mobility, reflecting longer-term structural changes across pandemic periods. Additionally, is_holiday and lockdown_level demonstrate relatively high gain in specific categories such as Workplaces and Transit Stations, indicating strong but localized effects associated with policy interventions or special periods.

In the Cover dimension (bottom panel), which represents the relative proportion of samples influenced by feature splits, temporal variables including avg_all_mobility, year, week, and day consistently affect a large share of observations across all mobility categories. The StringencyIndex and lockdown_level exhibit moderate coverage, particularly in work and travel related categories. Notably, day_of_week shows high coverage in Parks and Transit Stations, highlighting systematic differences in mobility behavior between weekdays and weekends.

4. Finally, to enhance transparency and reproducibility of the machine learning implementation, I strongly recommend that the authors provide the full code or scripts used for data preprocessing, model training, and evaluation. Making these scripts available—either as supplementary material or via a public repository (e.g., GitHub)—would allow other researchers to replicate the results and verify the methodology

Response :

Thank you for your comments, which help make our work beneficial to other researchers and increase the transparency and reproducibility of our work. In this regard, we have made our full code, data preprocessing, model training and evaluation available on GitHub at the following address: https://github.com/apisitgo/COVID19-Mobility-Forecasting-TH

---

## [Decision Letter · Decision Letter 1]

2 Mar 2026

Dear Dr. Rasmequan,

We look forward to receiving your revised manuscript.

Kind regards,

Jie Zhang

Academic Editor

PLOS One

Journal Requirements:

Reviewers' comments:

Reviewer's Responses to Questions

**Comments to the Author**

Reviewer #1: All comments have been addressed

2. Is the manuscript technically sound, and do the data support the conclusions?

Reviewer #1: Yes

3. Has the statistical analysis been performed appropriately and rigorously?

Reviewer #1: Yes

4. Have the authors made all data underlying the findings in their manuscript fully available?

Reviewer #1: Yes

5. Is the manuscript presented in an intelligible fashion and written in standard English?

Reviewer #1: Yes

Reviewer #1: Overall, the revision substantially improves transparency and model comparison. However, several consistency fixes and editorial corrections are still needed before acceptance.

For example:

• Duplicate entry — Radečić (items 24 and 32) appears twice; please deduplicate.

• Source quality — Replace non‑scholarly/tutorial sources (e.g., Medium, Analytics Vidhya) with primary documentation or peer‑reviewed publications wherever possible.

**Do you want your identity to be public for this peer review?** For information about this choice, including consent withdrawal, please see our For information about this choice, including consent withdrawal, please see our Privacy Policy .

Reviewer #1: No

---

## [Author Response · Author response to Decision Letter 2]

5 Mar 2026

Part 2 : Response to Reviewers:

1. If the authors have adequately addressed your comments raised in a previous round of review and you feel that this manuscript is now acceptable for publication, you may indicate that here to bypass the “Comments to the Author” section, enter your conflict of interest statement in the “Confidential to Editor” section, and submit your "Accept" recommendation.

Reviewer #1: All comments have been addressed

Response :

We appreciate the reviewer’s positive assessment. No changes were made to this section of the manuscript.

2. Is the manuscript technically sound, and do the data support the conclusions?

Reviewer #1: Yes

Response :

We appreciate the reviewer’s positive assessment. No changes were made to this section of the manuscript.

3. Has the statistical analysis been performed appropriately and rigorously?

Reviewer #1: Yes

Response :

We appreciate the reviewer’s positive assessment. No changes were made to this section of the manuscript.

4. Have the authors made all data underlying the findings in their manuscript fully available?

Reviewer #1: Yes

Response :

We appreciate the reviewer’s positive assessment. No changes were made to this section of the manuscript.

5. Is the manuscript presented in an intelligible fashion and written in standard English?

Reviewer #1: Yes

Response :

We appreciate the reviewer’s positive assessment. No changes were made to this section of the manuscript.

6. Review Comments to the Author

Reviewer #1: Overall, the revision substantially improves transparency and model comparison. However, several consistency fixes and editorial corrections are still needed before acceptance.

For example:

• Duplicate entry — Radečić (items 24 and 32) appears twice; please deduplicate.

• Source quality — Replace non‑scholarly/tutorial sources (e.g., Medium, Analytics Vidhya) with primary documentation or peer‑reviewed publications wherever possible.

Response :

We appreciate the reviewer’s constructive comments. We carefully reviewed the reference list by updated them with scholarly source and removed those duplication references as follows:

1. To improve source quality, non-scholarly/tutorial sources were replaced with peer-reviewed primary references:

1.1 Reference [24] was replaced with: Bergmeir C, Hyndman RJ, Koo B. A note on the validity of cross-validation for evaluating autoregressive time series prediction. Comput Stat Data Anal. 2018;120:70–83. https://doi.org/10.1016/j.csda.2017.11.003

1.2 Reference [25] was replaced with: Chen T, Guestrin C. XGBoost: A scalable tree boosting system. Proc 22nd ACM SIGKDD Int Conf Knowledge Discovery and Data Mining. 2016:785–794. https://doi.org/10.1145/2939672.2939785

2. Those non-scholarly and duplicate references were removed:

2.1 Former Reference [32] has been removed.

2.2 Former Reference [33] has been removed

3. The in-text citations in Section 3 (Results and Discussion) were updated accordingly, with former references [32, 33] has been replaced with [24, 25].

7. PLOS authors have the option to publish the peer review history of their article (what does this mean?). If published, this will include your full peer review and any attached files.

Do you want your identity to be public for this peer review? For information about this choice, including consent withdrawal, please see our Privacy Policy.

Reviewer #1: No

Response :

Not applicable. This comment concerns reviewer privacy preferences and does not require manuscript modification.

---

## [Editor Report · Decision Letter 2]

9 Mar 2026

Forecasting Thailand’s mobility trends using Feature Engineered XGBoost for pandemic crisis movement management

PONE-D-25-63776R2

Dear Dr. Rasmequan,

We’re pleased to inform you that your manuscript has been judged scientifically suitable for publication and will be formally accepted for publication once it meets all outstanding technical requirements.

Kind regards,

Jie Zhang

Academic Editor

PLOS One
---

## [Editor Report · Acceptance letter]

PONE-D-25-63776R2

PLOS One

Dear Dr. Rasmequan,

I'm pleased to inform you that your manuscript has been deemed suitable for publication in PLOS One. Congratulations! Your manuscript is now being handed over to our production team.

Kind regards,

on behalf of

Dr. Jie Zhang

Academic Editor

PLOS One